# Endothelial TRPV1 as an Emerging Molecular Target to Promote Therapeutic Angiogenesis

**DOI:** 10.3390/cells9061341

**Published:** 2020-05-27

**Authors:** Sharon Negri, Pawan Faris, Vittorio Rosti, Maria Rosa Antognazza, Francesco Lodola, Francesco Moccia

**Affiliations:** 1Laboratory of General Physiology, Department of Biology and Biotechnology “L. Spallanzani”, University of Pavia, 27100 Pavia, Italy; sharon.negri01@universitadipavia.it (S.N.); faris.pawan@unipv.it (P.F.); 2Center for the Study of Myelofibrosis, Laboratory of Biochemistry, Biotechnology and Advanced Diagnosis, IRCCS Policlinico San Matteo Foundation, 27100 Pavia, Italy; v.rosti@smatteo.pv.it; 3Center for Nano Science and Technology @PoliMi, Istituto Italiano di Tecnologia, via Pascoli 70/3, 20133 Milano, Italy; mariarosa.antognazza@iit.it (M.R.A.); francesco.lodola@iit.it (F.L.)

**Keywords:** therapeutic angiogenesis, Ca^2+^ signaling, TRPV1, vascular endothelial cells, endothelial colony forming cells, erythropoietin, simvastatin, evodiamine, photostimulation, organic semiconductors

## Abstract

Therapeutic angiogenesis represents an emerging strategy to treat ischemic diseases by stimulating blood vessel growth to rescue local blood perfusion. Therefore, injured microvasculature may be repaired by stimulating resident endothelial cells or circulating endothelial colony forming cells (ECFCs) or by autologous cell-based therapy. Endothelial Ca^2+^ signals represent a crucial player in angiogenesis and vasculogenesis; indeed, several angiogenic stimuli induce neovessel formation through an increase in intracellular Ca^2+^ concentration. Several members of the Transient Receptor Potential (TRP) channel superfamily are expressed and mediate Ca^2+^-dependent functions in vascular endothelial cells and in ECFCs, the only known truly endothelial precursor. TRP Vanilloid 1 (TRPV1), a polymodal cation channel, is emerging as an important player in endothelial cell migration, proliferation, and tubulogenesis, through the integration of several chemical stimuli. Herein, we first summarize TRPV1 structure and gating mechanisms. Next, we illustrate the physiological roles of TRPV1 in vascular endothelium, focusing our attention on how endothelial TRPV1 promotes angiogenesis. In particular, we describe a recent strategy to stimulate TRPV1-mediated pro-angiogenic activity in ECFCs, in the presence of a photosensitive conjugated polymer. Taken together, these observations suggest that TRPV1 represents a useful target in the treatment of ischemic diseases.

## 1. Introduction

The interior of blood vessels is lined by endothelial cells, which integrate chemical and physical stimuli deriving from circulating blood and surrounding tissues to redirect blood flow according to local energy requirements [1,2]. Vascular endothelial cells are activated by the local reduction in O_2_ tension, which follows an ischemic insult to undergo angiogenesis, i.e., the new blood vessel sprouting from adjoining capillaries, thereby attempting to compensate the vascular damage [3]. Additionally, the replacement of injured/apoptotic/senescent vascular endothelial cells may be accomplished by a population of circulating endothelial precursors physically engrafted within neovessels, known as endothelial colony forming cells (ECFCs), which display high proliferative potential and the capability to produce paracrine pro-angiogenic instructions [4,5,6]. Therefore, it has been suggested that ECFCs interact with mature endothelial cells, to maintain vascular integrity and function throughout postnatal life [4,5,6]. While excessive vascularization is associated with tumorigenesis, intraocular and inflammatory diseases, and pulmonary arterial hypertension (PAH) [7,8], insufficient vessel growth or impaired blood flow due to vessel obstruction may lead to severe ischemic disorders, including stroke, peripheral artery disease (PAD), pre-eclampsia, acute myocardial infarction (AMI), and ischemic retinopathies, as well as to neurodegeneration [9,10]. Therapeutic angiogenesis holds great promise for the treatment of ischemic disorders by harnessing multiple pharmacological, cellular, or bioengineering tools to reconstruct the damaged vascular network [11,12]. This strategy postulates that exogenous supplementation of trophic factors, such as growth factors or other bioactive compounds, or transplantation of autologous ECFCs, could foster revascularization in ischemic diseases [5,12,13,14]. Hence, shedding light on the signal transduction pathways driving angiogenesis and vasculogenesis might remarkably impact the regenerative efficiency of therapeutic angiogenesis [5,12,15].

Endothelial Ca^2+^ signals are known to play an important role in angiogenesis and vasculogenesis. Multiple angiogenic signals (e.g., vascular endothelial growth factor (VEGF), basic fibroblast growth factor (bFGF), platelet derived growth factor (PDGF), inflammatory mediators (e.g., thrombin, ATP, ADP, and acetylcholine), pleiotropic hormones (e.g., erythropoietin), and mechanical damage induce endothelial Ca^2+^ signals, thereby stimulating proliferation, migration, and tube formation, both in vivo and in vitro [16,17,18,19,20,21,22,23,24]. Likewise, VEGF, stromal derived factor-1α (SDF-1α), and the hypoxic human amniotic fluid secretome stimulate distinct intracellular Ca^2+^ signatures in ECFCs, to promote proliferation, in vitro tubulogenesis, migration, and neovessel formation [25,26,27,28,29]. Vascular endothelial cells and ECFCs rely on endogenous Ca^2+^ sources, such as endoplasmic reticulum (ER) and endolysosomal vesicles, and on the extracellular environment, to shape the Ca^2+^ response to pro-angiogenic cues [19,21,30]. Store-operated Ca^2+^ entry (SOCE), which is recruited upon depletion of the ER Ca^2+^ reservoir, represents the most common Ca^2+^ entry pathway upon which inositol triphosphate (InsP_3_)-synthesizing growth factors and chemokines converge [18,27,30,31,32,33]. An additional endothelial Ca^2+^ entry route is provided by the superfamily of non-selective Transient Receptor Potential (TRP) cation channels, which comprises 28 members [18,34,35]. Standard nomenclature grouped these 28 members in six subfamilies, based on their sequence homology: canonical (TRPC1-7), vanilloid (TRPV1-6), melastatin (TRPM1-8), ankyrin (TRPA1), mucolipin (TRPML1-3), and polycystin (TRPP). This last subfamily is composed by eight members, although only TRPP2, TRPP3, and TRPP5 display the function and structure of an ion channel [34,36,37]. TRP channels regulate several endothelial functions, mediating extracellular Ca^2+^ entry and recruiting an array of Ca^2+^-dependent decoders in response to chemical, thermal, and mechanical stimuli [18,34,35,38]. As recently reviewed in References [18,35], some TRP channel isoforms sustain both angiogenesis and vasculogenesis, including TRPC1 [39,40], TRPC3 [26,41], TRPV4 [42,43], and TRPV1 [42,44]. Interestingly, TRPV1 is rapidly emerging as an important player in endothelial cell migration, proliferation, and tubulogenesis through the integration of distinct environmental and intracellular cues [35,37]. It has recently been proposed that manipulating endothelial Ca^2+^ signaling could provide an alternative strategy to enhance the vasoreparative outcome of therapeutic angiogenesis and/or to improve the regenerative potential of endogenous endothelial and cardiac progenitors [15,28,45,46]. For instance, it has been suggested that the reparative phenotype of autologous ECFCs could be rejuvenated by transduction with TRPC3 [47], which remarkably increases the pro-angiogenic Ca^2+^ response to VEGF [26,48]. Based on this rationale, we herein discuss the hypothesis to exploit chemical and physical stimuli to target endothelial TRPV1 for regenerative purposes. First, we summarize current knowledge about the structure, gating mechanisms, and physiological roles of endothelial TRPV1. Then, we illustrate how TRPV1-mediated Ca^2+^ signals promote angiogenesis in vitro and neovessel formation in vivo in response to a wealth of small molecule agonists that could be harnessed in therapy. Finally, we describe a recent strategy to stimulate TRPV1-dependent pro-angiogenic activity in ECFCs in the presence of a photosensitive organic semiconductor. These evidences hint at TRPV1 as a promising molecular tool to favor post-ischemic revascularization.

## 2. Vasculogenesis and Angiogenesis

The development of the circulatory system requires the interplay of several finely regulated processes. There are three main mechanisms of vessel growth during embryonal and postnatal life: vasculogenesis, angiogenesis, and arteriogenesis (Figure 1) [3,9]. Vasculogenesis is defined as the de novo formation of vessels from aggregated endothelial precursors, while angiogenesis consists in neovessel formation from preexisting blood vessels and may be distinguished into sprouting angiogenesis and intussusceptive angiogenesis. The stabilization of these new sprouts by mural cells, e.g., pericytes and vascular smooth muscle cells (SMCs), is then termed arteriogenesis [9,49].

### 2.1. Vasculogenesis

Embryonal vasculogenesis represents one of the earliest events during organogenesis, as it starts already during gastrulation [50,51]. Vasculogenesis consists in the differentiation, expansion, and assembly of endothelial precursors into a functional blood vessel [49,52]. Vascular development in the embryo is initiated by mesodermal cells, also referred to as hemangioblasts, which migrate into the yolk sac, to form cellular aggregates known as blood islands. These are small clusters of round mesenchymal cells, which differentiate over time, thereby generating an outer layer of spindle-shaped endothelial precursors, i.e., angioblasts, and inner core of round primitive hematopoietic cells. Hence, angioblasts form sprouts and coalesce with adjacent blood islands to assembly into the early vascular plexus, which in turn generates primitive blood vessels (Figure 1A) [53,54].

### 2.2. Angiogenesis

Once the primitive vascular labyrinth has been built up, blood vessels are finely remodeled, to define tissues structures based on metabolic need, in a hierarchical tree-like network of arteries, arterioles, capillary beds, venules, and veins [19,49,55]. There are two angiogenic mechanisms: sprouting angiogenesis (Figure 1B) and intussusceptive angiogenesis (Figure 1C). Sprouting angiogenesis consists of neovessel formation from a preexisting capillary in response to a decrease in local oxygen tension [3,56]. It is characterized by different steps: vasodilation, increased vascular permeability, endothelial cell proliferation, and migration. In response to an angiogenic signal, e.g., VEGF, a leading tip cell is allowed to migrate in the cellular matrix after the degradation of basement membrane [57]. Thereafter, trailing endothelial stalk cells start to proliferate, determining sprout elongation and the trunk of new capillaries, without interrupting the connection with the parental vessel. Sprouts emerging from different vessels anastomose to constitute a functional vessel loop, allowing blood to flow, which also contributes to arterial–venous specification of endothelial cells [3,56]. Finally, the deposition of extracellular matrix and the recruitment of mural cells (e.g., pericytes and smooth muscle cells) leads to vessel maturation and stabilization. When the angiogenic signal is ceased, and upon restoration of the oxygen supply, endothelial cells return to a quiescent state, adopting their typical cobblestone-like shape [10,58]. The complexity and vascular surface of the capillary network can be further expanded by the insertion of a multitude of tissue pillars, according to a process observed for the first time in pulmonary capillary bed of neonatal rats [59] and referred to as intussusceptive (non-sprouting) angiogenesis [57].

### 2.3. Definition of Endothelial Progenitor Cells (EPCs) in the Adult: Myeloid Angiogenic Cells (MACs) and Endothelial Colony Forming Cells (ECFCs)

It has long been thought that sprouting angiogenesis represents the most important mechanism responsible for vessel growth and remodeling in postnatal life [3,9]. This dogma has been challenged in 1997 by Jeffrey Isner’s group, who reported, for the first time, the existence of circulating EPCs in peripheral blood of adult individuals [60]. A surge of investigations rapidly revealed that EPCs were mobilized from stem cell niches located in the endothelial intima of blood vessels and in bone marrow, to replace damaged/senescent endothelial cells and to rescue the vascular networks injured by an ischemic insult, thereby favoring tissue revascularization and rescuing local blood perfusion [6,18,61,62,63,64]. We refer the reader to several recent and exhaustive review articles which summarize the debate originated around the definition, classification, and characterization of the multiple EPC subtypes identified in peripheral circulation [6,65,66]. A recent consensus statement on nomenclature discouraged the indiscriminate use of the term EPC, as referred to the endothelial precursors circulating in the blood stream, in favor of a more stringent classification based on the cellular immunophenotype and pro-angiogenic activity [65]. Two intrinsically distinct lineages of cells encompass the multitude of EPC subtypes described in the literature: MACs, also known as “early” EPCs, and ECFCs, or late-outgrowth EPCs. MACs represent bone-marrow-derived hematopoietic cells which cooperate with ECFCs and support vascular repair and tissue regeneration through paracrine signaling [64]. ECFCs represent truly endothelial precursors, due to their high clonogenic potential and ability to physically engraft within emerging neovessels. In addition, ECFCs assemble into bidimensional tube networks in vitro, give rise to patent blood vessels, and anastomose with the host vasculature in vivo [67,68,69]. Furthermore, ECFCs sustain tissue revascularization by supporting the vasoreparative activity of other progenitor and stem cells, such as mesenchymal stem [70] and progenitor [4] cells, and by releasing a host of neurotrophic mediators, including growth factors [5], exosomes [71], and microvesicles [72]. A growing number of independent studies confirmed that ECFC transplantation was able to induce neovascularization and rescue blood perfusion in ischemic diseases, including PAD, AMI, ischemic brain disease, and retinopathy. It has therefore been proposed that ECFCs represent the most suitable cellular substrate to effect therapeutic angiogenesis in ischemic patients [12].

### 2.4. The Signaling Pathways of Angiogenesis and Vasculogenesis: The Role of Endothelial Ca^2+^ Signaling

As mentioned earlier, angiogenesis and vasculogenesis are finely regulated mechanisms [73]. Under physiological conditions, there is a balance between proangiogenic stimuli (e.g., VEGF, bFGF, PDGF, SDF-1-α, and angiopoietin-1 (ANG-1)/Tie2), and anti-angiogenic molecules (e.g., thrombospondin-1, endostatin, tumstatin, vasohibin, and C-X-C motif chemokine 10 (CXCL10) [3,19]. VEGF promotes angiogenesis synergistically with SDF-1α [74]. The tyrosine kinase receptor (TRK) VEGFR2, also denoted as kinase insert receptor (KDR), is the main isoform whereby VEGF signals to vascular endothelial cells and ECFCs [75,76]. Upon VEGF binding, VEGFR2 dimerizes and undergoes transphosphorylation, thereby recruiting downstream angiogenic pathways [40,75,77]. VEGF, for example, activates the following: (1) the phospholipase Cγ (PLCγ)-extracellular signal-regulated kinases 1/2 (ERK1/2) pathway, which is a crucial player during vascular development and in adult arteriogenesis (Figure 2A); (2) the phosphoinositide 3-kinases(PI3K)–protein kinase B (Akt)-mammalian target of rapamycin (mTOR) pathway, which underlies cell survival and the regulation of vasomotion and barrier function; (3) endothelial nitric oxide (NO) synthase (eNOS), thereby leading to NO release, which regulates endothelial cell proliferation, migration, and neovessel formation; and (4) Src and small GTPases pathways, which regulate cell shape, migration, and polarization [75] (Figure 2B). In particular, Src phosphorylates focal adhesion kinase (FAK), which in turn is a crucial player in the regulation of cell shape and adhesion [78]. Moreover, Src induces the disruption of adherens junctions and an increase in vascular permeability through vascular endothelial (VE)-cadherin phosphorylation [79]. Likewise, the G_i_-protein coupled receptor, CXCR4 mediates SDF-1α-induced endothelial motility by enrolling a wide array of intracellular signaling pathways, including PLCβ and PLCγ2, RAS-mitogen-activated protein kinase (MAPK), and PI3K-Akt-mTOR [27,80].

Another fundamental signal whereby VEGF and SDF-1α stimulate neovessel formation is an increase in intracellular Ca^2+^ concentration ([Ca^2+^]_i_) [18,22,27,32,35,81]. As mentioned in the Introduction, endothelial Ca^2+^ signals regulate all the crucial steps involved in angiogenesis and vasculogenesis, including decrease in permeability, proliferation, and migration [18,75]. A number of Ca^2+^-sensitive decoders mediate the pro-angiogenic outcome of endothelial Ca^2+^ signals: These include NFAT, NF-κB, RAS-MAPK, PI3K/Akt, eNOS, myosin light chain kinase (MLCK), Ca^2+^/calmodulin-dependent protein kinase II (CaMKII), and calpain [18,75]. VEGF and SDF-1α elicit a dynamic interplay between InsP_3_-dependent Ca^2+^ release and SOCE, to trigger multiple intracellular Ca^2+^ signatures [22,27,32,81]. Nevertheless, some TRP isoforms may support VEGF-induced extracellular Ca^2+^ entry to regulate angiogenesis and arteriogenesis [18,35]. For instance, TRPC3 mediates VEGF-induced Ca^2+^ entry in human dermal microvascular endothelial cells [82], HUVECs [83], and umbilical cord blood-derived ECFCs [26], whereas TRPM2 supports VEGF-induced extracellular Ca^2+^ entry in human pulmonary artery endothelial cells [84]. Additionally, endothelial TRP channels may stimulate neovessel formation by sensing changes in the local microenvironment. For instance, TRPM2 is also sensitive to reactive oxygen species (ROS) [84], whereas TRPC5 is stimulated by hypoxia, to induce sprouting angiogenesis [85]. In this view, TRPV1 is emerging as a novel regulator of angiogenesis by means of its ability to detect both pro-angiogenic cues (e.g., erythropoietin) and environmental inputs (e.g., heat and ROS).

## 3. TRPV1: Molecular Structure and Gating Mechanisms

TRPV1 represents the funding member of the TRPV subfamily. TRP channels are featured by six transmembrane (TM1-6) α-helix segments, cytosolic NH_2_-, and COOH-tails and present a cation-permeable region formed by a loop between TM5 and TM6 [35]. The NH_2_- and COOH- terminals present a wide variability inside the superfamily being featured by different functions and lengths. In some members, they may interact with regulatory domains of cytosolic proteins (e.g., kinases, cytoskeletal proteins, and Ca^2+^-dependent sensors) [36,86]. However, there are two highly conserved regions inside the TRP domain (TRP-box1 and TRP-box2) that limit a central sequence characterized by a high degree of variability [87]. Surprisingly, the TRP-box2 is absent in the TRPV subfamily, although this shows a TRP-like domain, whose α-helix configuration and role resemble those of canonical TRP domains [35]. Additionally, TRPV channels share multiple ankyrin repeat domains (ARDs) localized on NH_2_-terminal, which are responsible of several functions, such as cytoskeleton interactions and channel opening by membrane deformation and channel tetramerization [86]. TRPV subunits can assemble into homo- and heterotetramers, although it is not clear which combinations are present in the vascular system [34]. It has long been known that TRPV1 and TRPV4 subunits form heteromeric channel exclusively located on the plasma membrane [88]. Accordingly, TRPV4 was found to associate with TRPV1 into a functional heteromeric channel complex in mouse primary retinal microvascular endothelial cells [89]. Likewise, TRPV5 and TRPV6 may assemble into heteromeric complexes in the kidney and gastrointestinal tract, but they are yet to be reported in vascular endothelium [37,90]. Furthermore, TRPV subunits may physically interact with TRP channels from different subfamilies: For instance, TRPP2/TRPV4 [91], TRPC1/TRPV6 [92], TRPML3/TRPV5 [93], and TRPV4/TRPC6 [94] heteromeric combinations were reported in naïve cells from multiple tissues. Of note, TRPV4/TRPC1 can assemble into functional channels into vascular endothelial cells from different vascular districts [18], including human umbilical vein endothelial cells (HUVECs) [95], rabbit mesenteric artery endothelial cells [96], and mouse aortic endothelial cells [97], whereas TRPV4 has been shown to interact with TRPC1 and TRPP2, to form a flow-sensitive mechanosensor in naïve vascular endothelial cells [98].

### 3.1. TRPV1 Structure

TRPV1 has been the first TRP channel to be solved at cryo-EM resolution [99,100]. This analysis confirmed that each TRPV1 subunit consists of six TM domains arranged around a central ion conduction pathway contributed by TM5 and TM6 and the re-entrant pore loop between them. The highly conserved TRP domain forms an interfacial helix that is positioned at the COOH-terminus of TM6 and interacts with the pre-TM1 helix and the cytosolic linker between TM4 and TM5, thereby enabling channel ligands to allosterically modulate pore conformation. The outer mouth of TRPV1 tetramers is remarkably wide, although it is followed by a dual-gate channel pore presenting two restriction points at G643 (4.6 Å, the selectivity filter) and at I679 (5.3 Å, the hydrophobic lower gate) that are closed in the non-conducting state. However, TRPV1 activation requires major structural rearrangements in the ion conduction pore, including the re-entrant loop, the selectivity filter, and the lower gate. Accordingly, capsaicin, which is a potent activator of the channel, binds to a cytosolic pocket in proximity of the inner mouth and gates TRPV1 by pulling TM6 from adjacent subunits apart and dilating the lower gate, while the constriction at the selective filter is not relieved. Conversely, the spider toxin binds to an extracellular site near to the re-entrant pore and gates TRPV1 by tilting the pore helix away from the central axis of the ion conduction pathway, thereby widening the selectivity filter from 4.6 Å to 7.6 Å. At the same time, the channel pore is further expanded by disruption of the hydrogen bonds bridging the loops that follow TM5 and precede TM6 [99,100].

### 3.2. TRPV1: Biophysical Properties and Gating Mechanisms

The TRPV1 channel, also named vanilloid receptor 1 (VR1), is the most studied and characterized TRPV member [101]. TRPV1 is a non-selective cation channel with an outwardly rectifying current–voltage (I–V) relationship with a negative slope conductance region at membrane potentials more negative than −70 mV [102]. TRPV1 is featured by similar permeabilities to Na^+^ and K^+^, but it is considerably more permeable to Ca^2+^ and Mg^2+^ (P_Ca_/P_Na_ = 9.6; PMg/P_Na_ = 5), and displays a surprisingly high permeability to large cations [102,103]. Being mainly located on the plasma membrane, TRPV1 activation results in extracellular Ca^2+^ entry down the electrochemical gradient at physiological membrane potentials [102]. TRPV1 is a polymodal cation channel activated by several mechanical and chemical stimuli, such as extracellular protons, products of plants origin (e.g., capsaicin and gingerol), noxious heat (>42 °C), spider-derived vanillotoxins, and hydrogen sulphide (H_2_S) [18,101,104,105,106,107,108]. TRPV1 may also be gated by some endogenous agonists, including fatty acids conjugated with amines (e.g., *N*-arachidonylethanolamine (AEA, anandamide), *N*-arachidonoyldopamine (NADA), *N*-oleoylethanolamine (OLEA), *N*-arachidonolylserine, and various *N*-acyltaurines and *N*-acylsalsolinols), ATP, adenosine, polyamines (e.g., as spermines and spermidines), prostaglandins, and leukotriene B4 and pH < 5.9 [109,110,111,112,113,114]. Interestingly, an elegant study by Nilius’ group showed that TRPV1 is also sensitive to membrane depolarization and that heat and capsaicin can function as gating modifiers through voltage-to-current relationship shifts [115]. Indeed, at room temperature (22–23 °C), the voltage-dependent activation of TRPV1 requires a strong positive shift in the membrane potential (up to around +150 mV). Conversely, at higher temperatures, i.e., 40–45 °C, TRPV1 activation may already occur at −50 mV. This means that heat and capsaicin induce the channel to open at more physiological voltages [115]. It has been demonstrated that also extracellular protons act in a similar manner, thereby promoting TRPV1 activation at resting temperature, which provides is an additional explanation of TRPV1 involvement in inflammatory conditions [116]. Finally, TRPV1 undergoes desensitization, mainly through a Ca^2+^-dependent mechanism. Extracellular Ca^2+^ influx, mediated by TRPV1 itself, activates an inhibitory feedback signal by recruiting multiple signaling pathways. First, incoming Ca^2+^ could engage the Ca^2+^-dependent phosphatase, calcineurin, to counteract protein kinase C (PKC)- and protein kinase A (PKA)-dependent phosphorylation; second, Ca^2+^ could promote desensitization by binding to calmodulin [117].

## 4. The Physiological Role of TRPV1 in Vascular Endothelium

TRPV1 has been discovered in neuronal cells, being especially abundant in rat dorsal root ganglia (DRG) [101], and rapidly found also in trigeminal (TG) and nodose (NG) ganglia [107]. Several evidences until our days have shown a widely expression of TRPV1 also in non-neuronal cell, e.g., smooth muscular cells, keratinocytes, urothelium, liver, glial cells, mast cells, and macrophages [117,118,119]. Furthermore, TRPV1 is largely expressed in the intimal layer of blood vessels throughout peripheral vasculature, as reviewed in [119], and in umbilical cord blood (UCB)-derived [120] and circulating [40] ECFCs. The opening of TRPV1 in the plasma membrane causes the entry of extracellular Ca^2+^ down the electrochemical gradient, which therefore results in an increase in [Ca^2+^]_i_ [121,122,123]. There is no evidence that TRPV1-mediated Ca^2+^ influx results in regenerative ER-dependent Ca^2+^ release, following the recruitment of ryanodine-receptors through the process of Ca^2+^-induced release, as observed in rat dorsal root ganglion neurons [124]. TRPV1 expression has also been found in the ER in several cell types, including rat nociceptive neurons, [125] and breast [126] and prostate [127,128] cancer cell lines. Cytosolic localization of TRPV1 has been recently described in bovine retinal microvascular endothelial cells (RMECs) [89]. It is, however, still unclear whether intracellular TRPV1 contributes to endothelial Ca^2+^ signaling.

### 4.1. TRPV1 Mediates Endothelium-Dependent Vasodilation

The interrelationship between TRPV1 and vascular endothelium was uncovered for the first time in 1999, when Zygmunt and co-workers proposed TRPV1 involvement in the regulation of vascular tone. They administered the powerful vasodilator, anandamide, to multiple isolated vascular preparations, including rat hepatic and small mesenteric arteries and guinea-pig basilar artery [129]. Anandamide was in turn found to activate TRPV1 channels localized on perivascular sensory fibers. TRPV1 activation induced the release of the vasodilator neuropeptide, calcitonin gene-related peptide (CGRP), which induced vasorelaxation by promoting endothelium-dependent hyperpolarization (EDH) [129]. Lo and co-workers provided one of the first pieces of evidence in favor of the role of endothelial TRPV1 in the control of vascular tone [130]. They indeed demonstrated that the vasorelaxant effect of VOA (*N*-(4-*O*-[2-methoxy, phenoxyethylaminobutyl]-3-methoxy benzyl)-nonamide), a synthetized molecule deriving from capsaicin, was endothelium-dependent [130]. First, they administered VOA to normotensive Wistar-Kyoto rats (WKY) and spontaneously hypertensive rats (SHR), thereby demonstrating a remarkable drop in systemic blood pressure. Then, they found that VOA was able to induce vasorelaxation in isolated rat thoracic aorta and mesenteric artery precontracted by phenylephrine (PE) in a TRPV1-dependent manner. Subsequently, they showed that VOA was able to induce CGRP release, followed by EDH in isolated vessel rings and to stimulate NO release in HUVECs [130]. It was, therefore, concluded that VOA was able to activate TRPV1, thereby promoting vasodilation in a CGRP/EDH and NO-dependent manner [130]. Two years later, Huidobro-Toro’s group reported that, although ineffective on vasorelaxation, 100 nM anandamide was able to promote robust NO release in the rat mesenteric bed by activating endothelial TRPV1 channels [131]. Anandamide-induced NO production was mimicked by TRPV1 agonists (i.e., resineferatoxin and capsaicin) and attenuated by TRPV1 antagonists (i.e., 5-iodoresiniferatoxin or SB 366791) [131]. An additional evidence for TRPV1-mediated endothelium-dependent vasorelaxation was provided by Zhu’s group in 2010 [132]. Capsaicin-induced TRPV1 activation induced the Ca^2+^-dependent phosphorylation of eNOS at Ser^1177^ in a PKA-dependent manner [132]. Consistently, long-term stimulation of TRPV1 induced vasorelaxation and lowered systemic blood pressure in SHR mice, accompanied by CGRP secretion from perivascular nerves. In this view, TRPV1 may be a therapeutic target in hypertension therapy in high-risk populations [132].

### 4.2. TRPV1 Ameliorates Endothelial Dysfunction in Diabetes, Atherosclerosis, and Metabolic Syndrome

Subsequent studies investigated whether and how TRPV1 could be targeted to mitigate endothelial dysfunction [133,134,135,136,137,138]. Earlier evidences showed that heat-stress-induced TRPV1 activation promoted CGRP secretion, thereby attenuating endothelial dysfunction induced by LPC, in mouse mesenteric artery and HUVECs [137,138]. Subsequently, Sun and collaborators analyzed TRPV1 involvement in oxidative-stress-induced endothelial dysfunction in diabetes, which represents one of the major cardiovascular risk factors. TRPV1 expression and PKA phosphorylation, as well as NO production, were decreased in porcine iliac artery endothelial cells maintained under high-glucose conditions [133]. Nonetheless, capsaicin rescued TRPV1 expression, thereby inducing the upregulation of Uncoupling Protein 2 (UCP2) in a PKA-dependent manner [133]. UCP2, in turn, ameliorated endothelial dysfunction by reducing the production of mitochondria-derived ROS and increasing NO release [133]. In agreement with these observations, capsaicin administration promoted UCP2 expression, mitigated ROS production, and restored NO-mediated, endothelium-dependent vasorelaxation in diabetic mice in vivo [133]. These data, therefore, demonstrated that TRPV1 may protect from hyperglycemia-induced endothelial dysfunction through the PKA/UCP2 pathway. Consistently, capsaicin stimulated endothelial TRPV1 to recruit the downstream PKA/UCP signaling cascade and thus mitigate mitochondrial dysfunction and restore coronary vasodilation and myocardial perfusion in atherosclerosis-prone apoliprotein E-deficient mice [134]. An additional elegant study proposed an alternative TRPV1-mediated mechanism of protection in endothelial dysfunction [135]. This investigation analyzed the effect of rutaecarpine, a major quinazolinocarboline alkaloid harvested from the Chinese herbal medicine, Evodia, on endothelial injury induced by Ox-LDL, a major risk factor for atherosclerosis [135]. Administration of Ox-LDL for 24 h decreased HUVEC viability and NO production, enhanced LDH release, and induced monocyte adhesion to the endothelial monolayer. However, endothelial damage was attenuated by pretreatment with rutaecarpine in a dose-dependent manner. Notably, capsazepine, a competitive TRPV1 antagonist, attenuated the beneficial effects of rutaecarpine on endothelial function [135]. Likewise, TRPV1 could effectively be targeted to treat metabolic syndrome [136,139]. Capsaicin-induced TRPV1 activation was found to mediate vasorelaxation by inducing NO release and EDH activation in coronary microcirculation from lean Ossabaw miniature swine [140]. Nevertheless, endothelial TRPV1 expression was significantly downregulated in obese animals, thereby compromising capsaicin-induced vasorelaxation [140]. Thus, it has been proposed that dietary agonists of endothelial TRPV1 could be employed to rescue coronary vasodilation in the presence of multiple cardiovascular risk factors, including diabetes, atherosclerosis, and metabolic syndrome [136,140]. Additionally, TRPV1 may protect vascular endothelial cells exposed to pro-inflammatory conditions associated, for instance, to lipopolysaccharide (LPS) [133,141]. Wang and collaborators demonstrated that TRPV1-activation suppressed LPS-induced HUVEC inflammation through eNOS activation and NF-kB inhibition [142]. Additionally, a TRPV1 protective effect was also reported in renal microvascular endothelial cells harvested from salt-sensitive hypertensive mice [142].

### 4.3. TRPV1 Regulates the Blood–Brain Barrier (BBB) Integrity

The only vascular district where endothelial TRPV1 has been associated to the impairment of vascular homeostasis is brain microcirculation. TRPV1 is largely expressed in human and mouse cerebromicrovascular endothelial cells [143,144]. A preliminary report showed that TRPV1 activation by anandamide reduced permeability in a human model of BBB by likely stimulating CGRP release [129,145]. A recent investigation disclosed that the endothelial expression of TRPV1 was enhanced within the peri-contusional region in a mouse model of traumatic brain injury (TBI) [143]. Pharmacological blockade of TRPV1 with capsazepine attenuated edema formation by preventing the dismantling of the BBB in vivo [143]. Subsequent evaluation of an in vitro model of BBB by using the mouse bEnd.3 cerebromicrovascular endothelial cell line confirmed that mechanical stimulation (imposed by biaxial stretch injury) caused endothelial cell apoptosis following TRPV1 activation [143]. Taken together, these findings demonstrate that endothelial TRPV1 plays a protective role under several pathological conditions, such as inflammation, diabetes, and atherosclerosis, while its role in brain ischemic remains to be determined. It, therefore, appears that endothelial TRPV1 could represent a useful target to prevent or attenuate endothelial dysfunction across peripheral vasculature.

## 5. Stimulating TRPV1 to Promote Angiogenesis

A growing number of evidences showed the involvement of TRPV1 in the angiogenic process [18,34,35,37]. However, capsaicin and piperine, both TRPV1 agonists, were initially found to inhibit angiogenesis both in vitro and in vivo [146,147]. It could be envisaged that chronic TRPV1 activation could lead to an aberrant increase in [Ca^2+^]_i_, thereby inducing to Ca^2+^-dependent cell death [148]. This hypothesis is supported by the notion that capsaicin-induced prolonged Ca^2+^ entry promotes apoptosis in prostate cancer cells [128], breast cancer cells [149], and cervical tumor cells [150]. On the other hand, TRPV1 activation was shown to reduce the ischemia/reperfusion (IR) injury in several organs, including the heart, brain, lungs, and kidney [151]. For instance, it has been demonstrated that TRPV1 activation during IR injury enhances substance P and CGRP from perivascular nerves and thereby afford cardioprotection [151,152,153]. It has also been suggested that TRPV1 activation may be induced upon myocardial injury by the reduction in local pH or by the proteases that are liberated from injured cells and activate the signaling cascade downstream of protease-activated receptor 2 (PAR2) [151,154,155]. Similarly, pharmacological activation of TRPV1 was found to induce neuroprotective effects, including a decrease in infarct volume and improvement of neurofunctional score, against IR injury in the brain [156,157]. It is, however, still unclear whether TRPV1 activation mediates revascularization upon ischemic insults in the heart and brain [151].

### 5.1. TRPV1 Sustains Angiogenesis Independently on VEGF

A recent study demonstrated that TRPV1 is able to trigger an angiogenic activity, even in the absence of parallel pro-angiogenic signaling pathways by VEGF. It was indeed shown that resiniferatoxin (RTX), a selective TRPV1 agonist, induced an outwardly rectifying non-selective cation current in primary cultures of bovine RMECs [89]. RTX-induced membrane currents were blocked by the selective TRPV1 inhibitors, capsazepine and A784168; furthermore, pharmacological blockade of TRPV1 with capsazepine and A784168 prevented the assembly of a bidimensional tube network in Matrigel scaffolds in the presence of 20% porcine serum [89]. Notably, these treatments did not affect either VEGF-induced intracellular Ca^2+^ signals or tubular network formation. These findings clearly demonstrated that TRPV1 activation per se is able to deliver a pro-angiogenic Ca^2+^ signal to vascular endothelial cells [89]. In agreement with this observation, blocking TRPV1 with capsazepine and A784168 inhibited neovascularization in a mouse model of oxygen-induced retinopathy [89]. Moreover, while pharmacological inhibition of TRPV1 with the selective antagonist SBB366791 did not prevent VEGF-induced tube formation in HUVEC, genetic deletion of TRPV1 abrogated stromal neovascularization in an in vivo cornea after a cauterization injury at the central cornea [158]. Consistently, the expression of pro-inflammatory/angiogenic factors (i.e., transforming growth factor β1 (TGFβ1) and VEGF) was suppressed in TRPV1-knockout (TRPV1^−/−^) mice [158]. Thus, targeting TRPV1 could provide an effective strategy to treat severe ischemic disorders. Of note, proximity ligation assay showed that TRPV1 could assemble with TRPV4 and blocking TRPV4 with selective blockers (i.e., HC06 and RN1734) also inhibited angiogenesis both in vitro and in vivo [89].

### 5.2. TRPV1 Stimulates Re-Endothelialization Following Vascular Injury

Apart from these sporadic observations, TRPV1 is widely recognized as a main trigger of angiogenesis. For instance, a recent investigation showed that TRPV1 was able to promote re-endothelialization after vascular injury [159]. These authors exploited the scratch wound assay to demonstrate that capsaicin increased HUVEC migration and proliferation in vitro by increasing mitochondrial energy metabolism and mitofusin (Mfn2) expression. In agreement with a role for TRPV1-mediated Ca^2+^ entry, the angiogenic effect of the dietary agonist capsaicin was inhibited by the TRPV1 antagonist, capsazepine, and by the membrane permeable Ca^2+^ chelator, BAPTA. These data were then therapeutically translated in vivo by performing wire injury of carotid artery in TRPV1-knockout (TRPV1^−/−^) mice and their wild-type (WT) littermates. Capsaicin was found to accelerate re-endothelialization and reduced neointimal formation in WT mice, but not in TRPV1 knockout mice. The regenerative effect of capsaicin was attenuated by genetic silencing of Mtfn2 both in vitro and in vivo [159], although the mechanism whereby TRPV1-mediated Ca^2+^ entry controls Mtfn2 expression is currently unclear. Of note, these data are consistent with the hypothesis that endothelial Ca^2+^ signaling could be properly manipulated to prevent restenosis, i.e., the reduction in lumen diameter, which occurs after percutaneous coronary intervention [160,161].

### 5.3. Chemical Stimulation of TRPV1 Promotes Angiogenesis in a Ca^2+^-Dependent Manner

Subsequent investigations demonstrated that TRPV1 could also be targeted to stimulate angiogenesis in a Ca^2+^-dependent manner. As described earlier, the first approach consisted of evaluating the effect of dietary or herbal TRPV1 agonists, such as capsaicin and evodiamine, respectively. Evodiamine is the major bioactive compound extracted from one of the most popular Chinese herbs, known as Wu-Chu-Yu (*Evodiae fructus*; *Evodia rutaecarpa* Benth., Rutaceae) [162]. Ching et al. investigated TRPV1-mediated eNOS activation and NO-dependent angiogenesis both in vitro and in vivo [163]. They found that evodiamine and capsaicin induced eNOS activation by phosphorylation and consequent NO release: Both of these effects were inhibited by pharmacological (with capsazepine) and genetic (with a specific small interfering RNA, siRNA) silencing of TRPV1. Evodiamine-induced TRPV1 activation was then found to recruit the Ca^2+^-dependent PI3K/Akt/CaMKII signaling pathway, which turned out to be necessary for ligand-induced phosphorylation of both TRPV1 and eNOS (Figure 3) [163]. Indeed, TRPV1 served as a scaffold for the recruitment and formation of a supermolecular complex consisting also of Akt, CaMKII and eNOS, which favored eNOS phosphorylation and NO release (Figure 4). This signaling pathway was also detected in mouse aortic endothelial cells (MAECs), in which genetic deletion of TRPV1 still prevented evodiamine from recruiting the PI3K/Akt/CaMKII/eNOS signaling cascade [163]. Of note, intraperitoneally injected evodiamine increased eNOS, Akt, and CaMKII phosphorylation in WT, but not TRPV1^−/−^ mice. NO has long been known to promote neovascularization by stimulating both angiogenesis and vasculogenesis [136,164,165,166]. Consistently, the Matrigel plug assay confirmed that evodiamine promoted angiogenesis in vivo, although neovascularization was prevented in TRPV1^−/−^ and eNOS-deficient (eNOS^−/−^) mice [163]. Of note, atherosclerotic lesions were more pronounced in ApoE-knockout mice (ApoE^−/−^), a widely employed animal model for hyperlipidemia, upon further deletion of TRPV1 (ApoE^−/−^ TRPV1^−/−^). Likewise, evodiamine-induced phosphorylation of Akt, CaMKII, and eNOS was lower in ApoE^−/−^TRPV1^−/−^, as compared to TRPV1^−/−^ mice [163]. A subsequent report further showed that evodiamine and capsaicin recruited AMP-activated protein kinase (AMPK) to phosphorylate eNOS in a CaMKII-dependent manner (Figure 4) [167]. Indeed, evodiamine also induced AMPK phosphorylation, but this effect was inhibited by blocking TRPV1 with capsazepine and CaMKII with the selective inhibitor KN62 [167]. Finally, evodiamine-induced eNOS phosphorylation was strongly reduced by compound C, a specific AMPK blocker, by overexpressing a dominant negative AMPK (dnAMPK) in Primary Bovine Aortic Endothelial Cells (BAECs). In agreement with these observations, AMPK activity proved to be essential for the ligand-induced physical association between TRPV1 and eNOS. As expected, pharmacological (with capsazepine) and/or genetic (with dnAMPK) blockade of AMPK also inhibited evodiamine-induced tube formation in Matrigel scaffolds both in vitro and in vivo [167]. Of note, this investigation demonstrated, for the first time, that TRPV1 could be effectively targeted to stimulate therapeutic angiogenesis. Intraperitoneal injection of evodiamine promoted neovascularization in a mouse model of hindlimb ischemia in an AMPK-dependent manner. Moreover, evodiamine reduced atherosclerotic plaques and increased phosphorylation of AMPK and eNOS in ApoE^−/−^, but not ApoE^−/−^TRPV1^−/−^ mice [167]. These studies, therefore, strongly suggest that pharmacological stimulation of TRPV1 could represent an alternative strategy to induce therapeutic angiogenesis in ischemic tissues, even in the presence of established cardiovascular risk factors, e.g., hyperlipidemia.

The lipid-lowering drug simvastatin, which a 3-hydroxy-3-methylglutaryl-CoA reductase antagonist, is widely employed to reduce cholesterol biosynthesis and to reduce coronary artery disease events in subjects with or without diagnosed cardiovascular disease [168]. The beneficial effect of simvastatin on the cardiovascular system is strengthened by its documented ability to stimulate angiogenesis [169], trigger endothelium-dependent NO release [170,171], and improve left ventricular ejection fraction (LVEF) in heart failure [172]. Simvastatin has been shown to stimulate angiogenesis by causing VEGF expression following hypoxia-inducible factor-1α (HIF-1α) upregulation in vascular endothelial cells [169]. Furthermore, a recent report showed that simvastatin could also serve as TRPV1 agonist, thereby inducing NO release and NO-dependent angiogenesis [173]. Simvastatin (as well as lovastatin) was found to induce a rapid and prolonged (up to 240 min) increase in [Ca^2+^]_i_ in Human Mammary Epithelial Cells (HMECs). The Ca^2+^ response to simvastatin was inhibited by two structurally unrelated TRPV1 inhibitors, such as capsazepine and SB366791; furthermore, simvastatin-induced intracellular Ca^2+^ signals were potentiated in HEK293 cells overexpressing a full-length TRPV1 WT, while it was inhibited by transducing the cells with TRPV1-Y671K (a TRPV1 mutant which is not able to conduct Ca^2+^) [173]. Similar to evodiamine, simvastatin stimulated TRPV1 to induce the Ca^2+^-dependent assembly of the TRPV1/Akt/CaMKII/AMPK/eNOS complex (Figure 4), thereby inducing eNOS phosphorylation, NO release, and NO-dependent tube formation in vitro [173]. Furthermore, simvastatin was able to promote neovascularization also in Matrigel plugs in vivo. Of note, simvastatin-induced NO production, tube formation in vitro, and neovascularization in vivo were inhibited by pharmacological (with capsazepine) or genetic (TRPV1^−/−^ mice) blockade of TRPV1 [173]. In agreement with previous observations [169], NO may both induce HIF-1α expression [174] and stabilize HIF-1α activity [175]. It thus appears that TRPV1 activation represents an additional mechanism whereby statins (such as simvastatin and possibly lovastatin) exert a beneficial outcome on the cardiovascular system. Future pre-clinical studies are mandatory to assess whether simvastatin-induced TRPV1 activation can effectively be employed to stimulate therapeutic angiogenesis. Moreover, 14,15-Epoxyeicosatrienoic acid (14,15-EET) is a cytochrome P450 epoxygenase (CYP)-derived metabolite of arachidonic acid, which stimulates endothelial hyperpolarization [136], activates eNOS to induce NO release [176] (Figure 3), and stimulates endothelial cell migration and tube formation [177]. Pharmacological blockade of soluble epoxide hydroxylase (sHE), which is responsible for the rapid degradation of 14,15-EET into a less-active metabolite, has been put forward as an alternative mechanism to treat multiple cardiovascular disorders, such as atherosclerosis, hyperlipidemia, and hypertension [178]. A recent study demonstrated that 14,15-EET induced a rapid and prolonged (up to 60 min) increase in [Ca^2+^]_i_ that disappeared in the absence of extracellular Ca^2+^ and upon pharmacological blockade of TRPV1 with capsazepine or SB366791 in HMEC (Figure 3) [121]. Similar to simvastatin, 14,15-EET-induced Ca^2+^ entry was, respectively, boosted or inhibited in HEK293 cells overexpressing a full-length TRPV1 WT or the TRPV1-Y671K with defective Ca^2+^ permeability. Thus, TRPV1 could be activated also by 14,15-EET, although through the intermediation of a GPCR [121], which is likely to be GRP40 [179]. Indeed, 14,15-EET-induced extracellular Ca^2+^ entry was blocked by the EET receptor blocker, 14,15-epoxyeicosa-5(Z)-enoic acid (14,15-EE5ZE), and a recent investigation disclosed that GRP40 activation resulted in a Ca^2+^ response similar to that evoked by 14,15-EET [179]. As reported above for evodiamine and simvastatin, 14.15-EET-induced extracellular Ca^2+^ entry then recruited eNOS to stimulate NO release and favor bidimensional tube formation in vitro and neovessel growth in Matrigel plugs in vivo (Figure 4) [121]. These findings, therefore, suggest that targeting sHE to promote 14,15-EET accumulation, thereby stimulating TRPV1-dependent pro-angiogenic Ca^2+^ signals, represents another promising strategy to treat ischemic disorders.

An additional approach to stimulate TRPV1-dependent angiogenesis is through dietary supplementation of epigallocatechin-3-gallate (EGCG) [122]. EGCG is one of the major effective components of green tea, by accounting for 50%–80% of its catechins [180]. Recent evidence suggested that green tea consumption could promote angiogenesis in the ischemic brain [181], whereas epidemiological studies and meta-analyses revealed that it could also confer protection against cardiovascular disease by stimulating endothelium-dependent NO release [182]. A recent investigation showed that EGCG was able to stimulate TRPV1 to induce NO release in BAECs [122]. EGCG induced a rapid and sustained (up to 240 min) elevation in [Ca^2+^]_i_ that disappeared in the absence of extracellular Ca^2+^ and upon pharmacological blockade of TRPV1 with capsazepine or SB366791 (Figure 3). Moreover, EGCG-induced extracellular was, respectively, boosted or inhibited in HEK293 cells overexpressing a full-length TRPV1 WT or the TRPV1-Y671K with defective Ca^2+^ permeability [122]. Subsequent analysis confirmed that EGCG also stimulated TRPV1 to recruit the Ca^2+^-dependent Akt-CaMKII-AMPK signaling cascade, thereby leading to eNOS phosphorylation and NO release (Figure 4). Consistently, EGCG induced BAEC proliferation, migration, and tube formation through TRPV1 activation and NO release. Furthermore, EGCG promoted neovascularization in Matrigel plugs implanted in WT, but not TRPV1^−/−^ mice [122]. These findings support the notion that tea green supplementation could be beneficial for cardiovascular patients, especially those affected by ischemic disorders. Finally, TRPV1 has been shown to support the pro-angiogenic effect of the erythroid growth factor, erythropoietin [123,183]. Erythropoietin stimulates an angiogenic behavior, e.g., proliferation, migration, and tube formation, both in vascular endothelial cells [184,185] and in circulating ECFCs; furthermore, the vasoreparative potential of ECFCs is enhanced by pretreatment with erythropoietin in an array of ischemic disorders [12], including hindlimb ischemia [186], cerebral ischemia [187], and retinopathy [89]. Notably, a recent investigation revealed that erythropoietin-induced angiogenesis requires TRPV1-mediated Ca^2+^ entry in BAECs [123]. By using the same approach previously described for simvastatin, 14,15-EET, and EGCG, e.g., pharmacological blockade with capsazepine and SBB366791 and genetically-mediated potentiation or inhibition with the full-length TRPV1 WT or the TRPV1-Y671K mutant defecting in Ca^2+^ permeation, EPO was found a sustained (up to 240 min) increase in [Ca^2+^]_i_ [123]. Erythropoietin-induced intracellular Ca^2+^ signaling through TRPV1, in turn, recruited the Akt/AMPK signaling cascade to induce eNOS phosphorylation and NO release (Figure 4), thereby promoting tube formation in vitro and neovessel growth in Matrigel plugs implanted in WT, but not TRPV1^−/−^ mice [123]. Of note, erythropoietin did not activate TRPV1 directly, but through the engagement of PLCγ1; indeed, pharmacological (with the selective blocker U73122 [188]) and genetic (via a selective siRNA) blockade of PLCγ1 prevented the Ca^2+^ response to erythropoietin by attenuating TRPV1 phosphorylation [123]. These findings, therefore, further support the notion that recombinant erythropoietin could be administrated to treat ischemic disorders and demonstrate that TRPV1 plays a pivotal role in the recruitment of the pro-angiogenic signaling pathways.

### 5.4. Is There a Role for TRPV1 in Heat-Induced Angiogenesis?

TRPV channels could confer temperature-sensing properties to vascular endothelial cells. For instance, TRPV4 may be activate by a temperature increase (from 19 to 38 °C) in mouse aortic endothelial cells [189]. A more recent paper suggested that TRPV1 could also sense heat increases to over 40 °C in human corneal endothelial cells [190]. The temperature sensibility of endothelial TRPV channels has been related to changes in NO release, which in turn regulates vascular tone and endothelial permeability [189,191]. An angiogenic stimulus could be triggered by whole-body hyperthermia (41.5–42.5 °C for 15 min, from a resting temperature of 37 °C) in rat cardiac myocardium. Likewise, far-infrared dry sauna (39 °C for 15 min followed by 34 °C for 20 min once daily for four weeks), also known as Waon therapy, promoted an increase in myocardial capillary density and attenuated cardiac hypertrophy in hypertensive rats [192]. The same approach was found to induce myocardial revascularization and attenuate maladaptive cardiac remodeling in a murine model of AMI [193]. Finally, near-infrared red exposure promoted capillary growth in the skin of human subjects, while locally increasing the temperature from 37 to 42 °C [194]. Of note, these procedures brought the whole-body and local temperature beyond the threshold for TRPV1 activation. The use of TRPV1^−/−^ mice or the effect of Waon therapy in the presence of TRPV1 blockers are, however, required to assess TRPV1 involvement to the angiogenic switch observed in response to temperature increases.

## 6. TRPV1 Controls the Angiogenic Activity in ECFCs

A growing number of studies revealed the crucial role of Ca^2+^ signaling in circulating and umbilical-cord-blood-derived ECFCs [195,196]. Intracellular Ca^2+^ signals lie, indeed, at the center of an intricate network of signaling cascades that finely tune ECFC proliferation, migration, and neovessel formation [27,28,32,81]. For instance, VEGF elicits intracellular Ca^2+^ oscillations and promotes angiogenesis in circulating [32] and umbilical-cord-blood-derived ECFCs [26], whereas SDF-1α stimulates ECFC migration through a biphasic Ca^2+^ signal [27]. VEGF and SDF-1α trigger distinct intracellular Ca^2+^ signatures by inducing a concerted interplay between InsP_3_-induced ER Ca^2+^ release and SOCE [27,32]. However, TRPC3 has been shown to initiate the spiking Ca^2+^ response to VEGF in umbilical-cord-blood-derived ECFCs [26]. In addition, arachidonic acid, which is massively released in circulation by ischemic tissues [15], may also stimulate ECFCs to proliferate by activating TRPV4 followed by NO release [197]. The control of stem and progenitor cell fate is emerging as a compelling urgency for regenerative medicine [198]. In this regard, manipulating the signaling pathways that drive ECFC proliferation, migration, differentiation, and tubulogenesis could improve their regenerative outcome in vivo [6,12]. It has been proposed that Ca^2+^ signaling could be targeted to boost the regenerative potential of autologous ECFCs for regenerative purposes [15,47]. It has recently been demonstrated that TRPV1 is expressed and mediates extracellular Ca^2+^ entry in ECFCs [44,120]. Unlike bovine RMECs [89], TRPV1 is unlikely to interact with TRPV4 as the Ca^2+^ response to a specific TRPV4 agonist (GSK1016790A) was not sensitive to capsazepine [43]. Of note, TRPV1 controls ECFCs’ angiogenic activity both in a Ca^2+^-dependent and independent manner [44,120]; in particular, Lodola et al. provided the proof-of-concept that the optoceutical manipulation of TRPV1 is able to stimulate in vitro angiogenic [44], thereby paving the way for the therapeutic translation of this approach.

### 6.1. TRPV1 may Stimulate ECFC Proliferation in a Ca^2+^-Independent Manner

Anandamide is an endogenous endocannabinoid which may confer protection against multiple cardiovascular disorders, including hypertension and ischemia/reperfusion injury [199]. Earlier work showed that anandamide was not able per se to induce vasorelaxation in isolated rat mesenteric arteries, but it induced a rapid and transient NO production in endothelial cells that was inhibited by TRPV1 antagonists (e.g., 5-iodoresiniferatoxin, SB366791 or capsazepine) and mimicked by the TRPV1 agonists, RTX, and capsaicin [131]. As shown above, NO mediates TRPV1-dependent angiogenesis. Of note, a recent report demonstrated that anandamide entered into a HUVEC-derived cell line, i.e., EA.h926) and in ECFCs, through TRPV1, thereby stimulating proliferation and tube formation in a Ca^2+^-independent manner [120] (Figure 3). Indeed, anandamide uptake was inhibited by pharmacological (with SB366791) or genetic (via a siTRPV1) abrogation of TRPV1, while it was enhanced upon TRPV1 overexpression. Furthermore, capsaicin significantly attenuated anandamide uptake, which hints at a form of competition for the same binding site and is further consistent with TRPV1 engagement. Of note, anandamide uptake was not affected by removal of extracellular Ca^2+^, which seemingly ruled out the involvement of intracellular Ca^2+^ signaling in the downstream effects of TRPV1 [120]. Nevertheless, anandamide induced proliferation and tube formation in ECFCs, but this pro-angiogenic effect was abrogated by interfering with TRPV1 activity (with capsazepine) or expression (via a siTRPV1). It should be recalled that it has already been shown that TRP channels may support angiogenesis in a flux-independent manner [18]. For instance, genetic deletion of TRPC1 and TRPC4 inhibited proliferation, although it did not affect SOCE, in HUVECs [31]. Whatever the underlying mechanism, however, these preliminary data demonstrate that TRPV1 stimulates ECFC proliferation.

### 6.2. Gene-Less Opto-Stimulation of TRPV1 Leads to In Vitro Modulation of ECFC Fate

Pharmacological modulation of TRPV1 activity for regenerative purposes may suffer of important drawbacks that limit their applicative potential. More specifically, the injection of TRPV1 agonists, e.g., simvastatin, evodiamine, and erythropoietin, has limited spatial resolution and is often irreversible and does not allow temporally precise control.

Use of light as a stimulation tool recently emerged as a valid alternative, since it allows targeting single cells or even cellular sub-compartments, providing unprecedented spatial and temporal resolution, lower invasiveness, and higher selectivity. However, since the vast majority of mammalian cells do not bear any specific sensitivity to light, several strategies have been proposed to transduce light excitation into a biological stimulus. Among them, optogenetics, which regards the use of the genetic modification of cells to express light-sensitive actuators, is currently considered one of the most powerful methods for in vitro optical stimulation; still, the need for viral gene transfer raise important concerns toward its future clinical applicability [200]. For this reason, gene-less opto-stimulation of living cells and tissues is a growing field of research at the border among biophotonics, materials science, and biology [201].

Light-sensitive organic semiconductors are emerging as highly promising materials in biotechnology, thanks to a series of key enabling characteristics [202]: (i) they efficiently absorb light in the visible spectral range and undergo charge photogeneration; (ii) they sustain both electronic and ionic charge transport; (iii) they are soft, conformable, and solution processable; (iv) they can be easily tuned to enable specific excitation, probing, and sensing capabilities; and, most importantly, (v) they are highly biocompatible. Interestingly, a reliable optical modulation of the cell activity mediated by conjugated polymers has been reported in several previous reports, in vitro, at the level of single cells [203,204,205,206,207,208], ex vivo [209], and in vivo, as evidenced by behavioral studies on both invertebrate and vertebrate models [210,211].

Three different photostimulation mechanisms that are active at the polymer/cell interface have been proposed so far [202]. These include the following: (i) the creation of an interface capacitance, i.e., of a localized electric field, possibly affecting the cell membrane potential [212]; (ii) thermally mediated effects, which can modify the cell membrane electrical/mechanical/physical properties of the nearby cell [207,213]; (iii) photoactivated reduction/oxidation reactions occurring at the interface, leading to a local variation of extracellular and/or intracellular pH and sizable production of ROS at non-toxic concentration [202,207,214], and intracellular calcium modulation [215,216].

Recently, conjugated polymer semiconductors have also been used to gain optical control of cell fate via TRPV1 activation [207]. Lodola and colleagues demonstrated that rr-P3HT (regioregular poly(3-hexylthiophene))-mediated optical excitation during the first steps of ECFCs growth led to a robust increase of both proliferation and lumen formation in vitro (Figure 5A–C). Experimental evidences corroborated the biophysical scenario that optical excitation was able to stimulate ECFCs through TRPV1-mediated extracellular Ca^2+^ entry. This caused the downstream activation of Ca^2+^-dependent transcriptional factor NF-κB, thus leading to the expression of multiple pro-angiogenic genes that converted the initial optical excitation into an angiogenic response (Figure 5D). Of note, NF-κB is the Ca^2+^-dependent decoder that translates VEGF-induced intracellular Ca^2+^ oscillations into a pro-angiogenic output [32]. Finally, photostimulation-induced membrane depolarization was not sensitive to the pharmacological blockade of TRPV4, again arguing against the physical/functional interaction between TRPV1 and TRPV4. For the sake of completeness, the authors shone light also on the phototransduction mechanism demonstrating how the excitation of the TRPV1 channel through direct photothermal transduction seems not the predominant process, leading to enhanced tubulogenesis, thus pointing out how polymer surface photocatalytic activity played a major role in the phenomenon. Indeed, as mentioned earlier, ROS may activate TRPV1. It has, therefore, been suggested that optical stimulation of P3HT polymer induces the production of singlets and charge states, which in turn reduce oxygen dissolved in the medium to form superperoxide. Furthermore, superperoxide leads to the generation of hydrogen peroxide (H_2_O_2_), which can permeate the plasma membrane through specific aquaporins (e.g., aquaporin 3) and thereby activate TRPV1 [44,207,217]. Overall, this work evidenced how the combined use of optical excitation and organic polymers technology can open interesting perspectives for the modulation of progenitor cells fate and, more generally, corroborated the idea that controlling the signaling pathways driving ECFC proliferation, differentiation, and tube formation may represent a reliable strategy to improve the regenerative outcome of therapeutic angiogenesis.

## 7. Conclusions

Therapeutic angiogenesis represents one of the most promising strategies to rescue local blood flow in ischemic organs, when pharmacological treatment is ineffective and surgical vascularization is not feasible. Manipulating the intracellular signaling pathways, which drive endothelial proliferation, migration, and neovessel formation, could be crucial to enhance the clinical outcome of regenerative medicine. Therefore, unravelling novel pro-angiogenic mediators will enrich the arsenal of signaling pathways that can be targeted for therapeutic purposes. The polymodal TRPV1 channel is emerging as a crucial regulator of angiogenesis, due to its ability to integrate multiple chemical and physical stimuli, ranging from dietary agonists (e.g., capsaicin and EGCG) and the arachidonic acid-derivative 14,15-EET to a moderate increase in microenvironmental temperature or ROS levels. Furthermore, recent work demonstrated that some clinically available drugs, such as simvastatin or recombinant erythropoietin, could be repurposed to stimulate angiogenesis by activating TRPV1-mediated extracellular Ca^2+^ entry. The available evidence hints at vascular endothelial cells as the final actuators of agonists-induced TRPV1-dependent Ca^2+^ signals. Nevertheless, the evidence that TRPV1 is also expressed and mediates angiogenesis in circulating ECFCs strongly suggests that systemic or local administration of these small molecule drugs might also stimulate vasculogenesis. In this regard, a novel approach demonstrated that optical stimulation of TRPV1 could stimulate ECFCs with unprecedented spatial and temporal resolution and with no need of viral infection. Future work is necessary to assess whether this approach is also useful to stimulate capillary endothelial cells and to design a therapeutically relevant approach. This requires, e.g., the design of a strategy to precisely deliver visible light to the injured tissue or to exploit an infrared-sensitive photosensitive conjugated polymer, as well as the production of biocompatible and photosensitive nanoparticles to be injected within ischemic organs. Additional issues that remain to be clarified to better exploit TRPV1 for regenerative purposes are the following. First, it should be assessed whether TRPV1 supports VEGF-induced endothelial Ca^2+^ signals in vascular districts other than bovine retina. Likewise, TRPV1 contribution to the pro-angiogenic effect of other growth factors, such as bFGF and PDGF, should be evaluated. Second, it should be evaluated whether the intracellular distribution of TRPV1 is restricted to bovine RMECs or is a common feature in vascular endothelial cells and ECFCs. If so, the contribution of this intracellular TRPV1 pool to endothelial Ca^2+^ signaling and its Ca^2+^-dependent interaction with InsP_3_- and/or ryanodine-receptors should be investigated. Third, future work will have to show whether pharmacological or optoceutical stimulation of TRPV1 leads to revascularization also in the ischemic brain/heart. This issue is quite relevant due to the increasing incidence of brain stroke and AMI in the aging population. Fourth, TRPV1 may assemble with another pro-angiogenic Ca^2+^ entry pathway, such as TRPC1 [97,218], in response to ER Ca^2+^ depletion. It is, however, still unclear whether this signaling complex stimulates angiogenesis, an issue that surely deserves future investigation.

## Figures and Tables

**Figure 1 cells-09-01341-f001:**
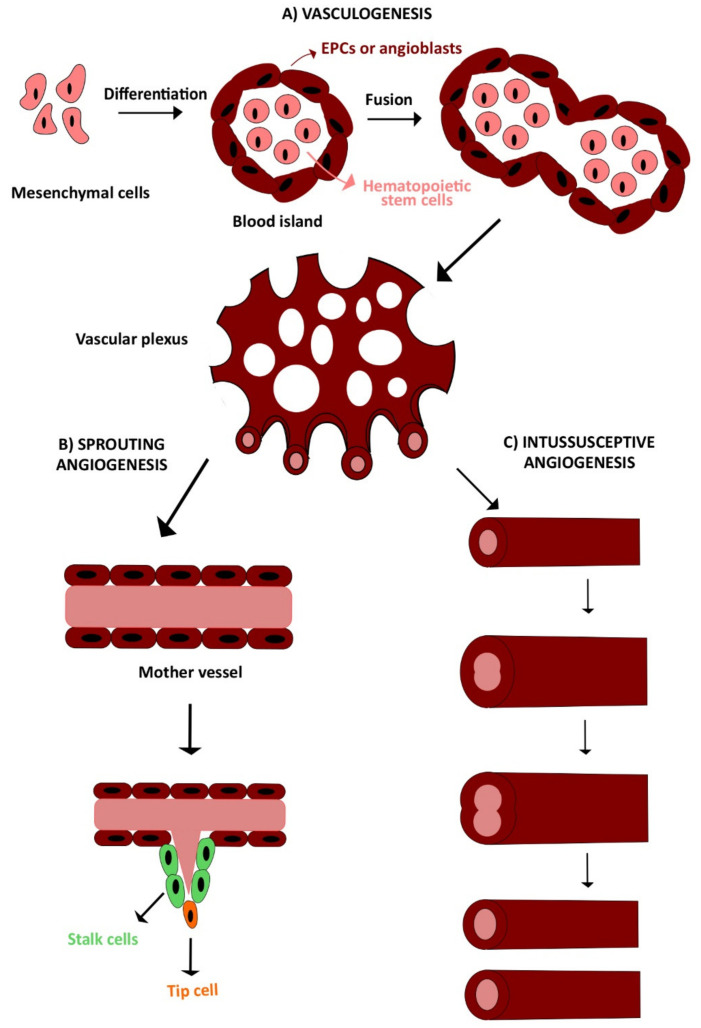
The two main processes involved in vascular development: vasculogenesis and angiogenesis. (**A**) Schematic representation of vasculogenesis, which consists of de novo vessel formation from aggregated endothelial precursors (EPCs or angioblasts) assembled within the blood islands. Thereafter, multiple blood islands fuse together into the early vascular plexus, which in turn generates primitive blood vessels. (**B**,**C**) Angiogenesis consists of neovessel formation from preexisting blood vessels in response to pro-angiogenic signals. Angiogenesis may occur through two different mechanisms: sprouting angiogenesis and intussusceptive angiogenesis. See text for detailed explanation.

**Figure 2 cells-09-01341-f002:**
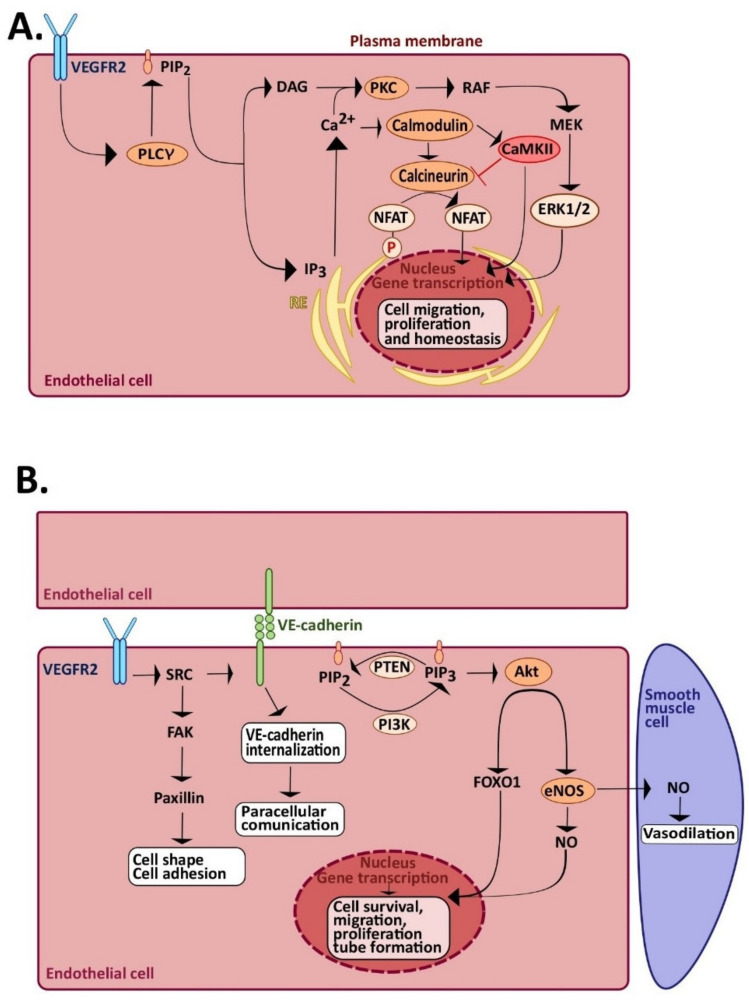
Role of endothelial Ca^2+^ signaling angiogenesis and vasculogenesis. (**A**) VEGF binds to VEGFR2 and induces receptor dimerization and transphosphorylation. Activation of VEGFR2, in turn, leads to InsP_3_-dependent Ca^2+^ release, followed by SOCE activation (not shown). The ensuing increase in [Ca^2+^]_i_ leads to the activation of Ca^2+^-dependent decoders, such as calmodulin, which in turn activates calcineurin and Ca^2+^-Calmodulin-dependent kinase II (CaMKII). Calcineurin dephosphorylates NFAT, thereby promoting its translocation into the nucleus, where it activates genes responsible for cell proliferation and migration. CaMKII, in turn, inhibits calcineurin activity, but stimulates angiogenesis by phosphorylating multiple targets (i.e., Akt and Src, not shown). In addition, the increase in [Ca^2+^]_i_ promotes cytosolic protein kinase C (PKC) relocation toward the plasma membrane. Herein, PKC is activated by DAG to stimulate the extracellular signal-regulated kinases 1/2 (ERK1/2) phosphorylation cascade. (**B**) Schematic representation of the Src and PI3K-Akt signaling pathways activated by VEGF. Once activated, VEGFR2 activates Src and subsequent downstream pathways involved in cell shape, adhesion, permeability, and survival. Moreover, VEGFR2 indirectly activates PI3K, either by Src or by VE-cadherin. PI3K generates phosphatidylinositol-3,4,5-trisphosphate (PIP_3_), which activates Akt, followed by eNOS activation and NO release. Finally, Akt activates FOXO1, which translocates into the nucleus, where, together with NO, induces transcription of genes involved in cell survival, migration, proliferation, and tube formation.

**Figure 3 cells-09-01341-f003:**
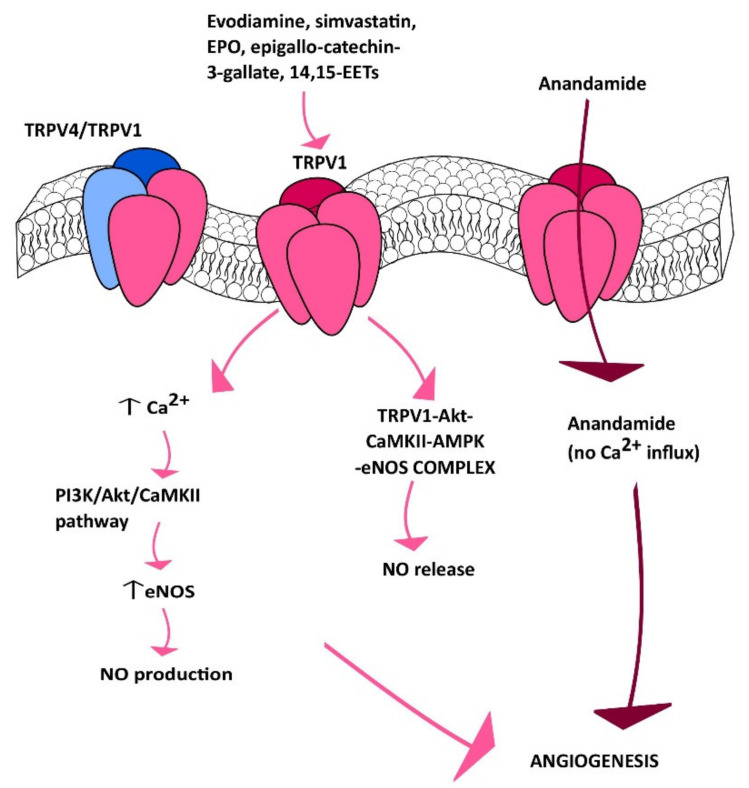
TRPV1 channel in angiogenesis. TRPV1 stimulates angiogenesis in response to evodiamine, simvastatin, EPO, epigallo-catechin-3-gallate, and 14,15-EETS in a Ca^2+^-dependent manner. Conversely, extracellular anandamide may enter through TRPV1, thereby stimulating angiogenesis in a Ca^2+^-independent manner.

**Figure 4 cells-09-01341-f004:**
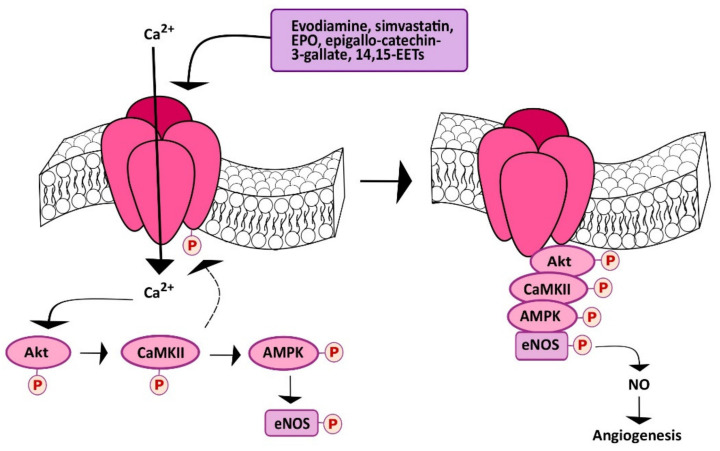
Proposed molecular mechanism of eNOS stimulation after TRPV1 activation. Activation of TRPV1 increases Ca^2+^ influx, which in turn activates PI3K/Akt/CaMKII signaling, leading to increased TRPV1 and eNOS phosphorylation. In addition, TRPV1 may serve as a scaffold for the formation of a complex comprising Akt, AMPK, CaMKII, and eNOS. Protein interactions seem to be important in eNOS activation and NO release.

**Figure 5 cells-09-01341-f005:**
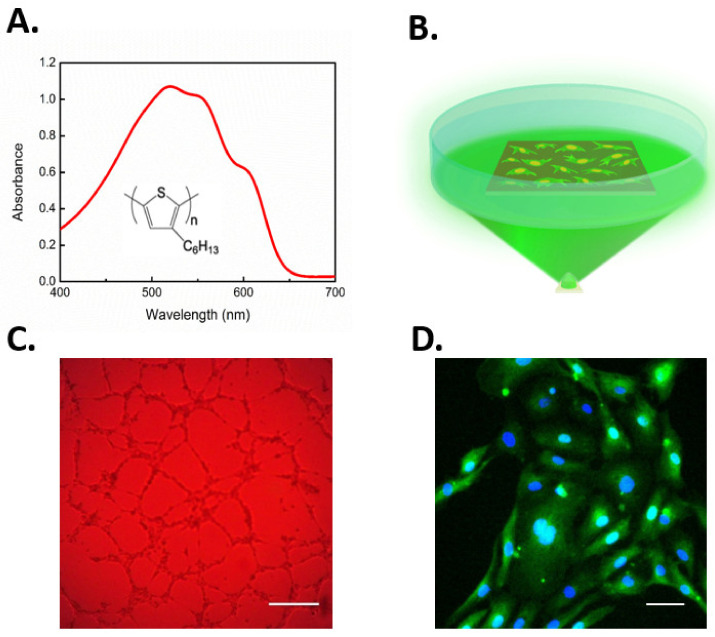
(**A**) The rr-P3HT chemical structure and optical absorption of the thin film. (**B**) Experimental setup and optical excitation protocol performed for evaluation of polymer-mediated cell photoexcitation effects on cell fate. (**C**) Representative image of in vitro network formation in ECFCs seeded on P3HT and subjected to optical excitation; scale bar: 250 µM. (**D**) Representative image of immunofluorescence staining, showing enhanced light-induced NF-κB nuclear translocation. Cell nuclei are detected by DAPI (blue) while cytoplasmic p65 NF-κB subunit with a secondary chicken anti-rabbit Alexa(488)-conjugated antibody (green). Scale bar: 50 µM. Figures adapted from [44,207].

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
