# Peer review of "Endothelial TRPV1 as an Emerging Molecular Target to Promote Therapeutic Angiogenesis"

_cells, 2020, doi:10.3390/cells9061341_

Round 1
Reviewer 1 Report
This review well summarized the physiological roles of TRPV1 in vascular endothelium, and how endothelial TRPV1 promotes angiogenesis. The authors first introduced angiogenesis and endothelial progenitors, and the structure, gating mechanisms of endothelial TRPV1. Then the authors described TRPV1-mediated Ca2+ signals promoted angiogenesis. Finally, a recent strategy to opto-stimulate TRPV1-dependent pro-angiogenic activity in ECFCs were highlighted.
In general, the review is well written. However, this review is unfocused. In particular in the first part, the authors introduced a lot about vasculogenesis and angiogenesis, EPCs, ECFCs, and calcium related signalings, which are not related to the theme of this review. I recommend to simplify this part. Other concerns including:
- Did the authors draw these figures? If not, the authors need to clarify where are these figures adapted from.
- Further perspective should be added in this review.
- What is the correlation between TRPV1 and calcium? How TRPV1 stimulate calcium signal in ECs should be summarized.
- The role of TRPV1 in ischemic disease, such as ischemic brain/heart should be illustrated.
Author Response
Dear Referee #1,
We thank you for the nice comments on our manuscript entitled: Endothelial TRPV1 as an emerging molecular target to promote therapeutic angiogenesis recently submitted for publication as a Review article in Cells.
We believe that your comments did improve the quality of the manuscript and we are pleased for the attention you paid to our work. We amended the manuscript according to your indications and addressed all the criticisms you raised. More specifically:
In general, the review is well written. However, this review is unfocused. In particular in the first part, the authors introduced a lot about vasculogenesis and angiogenesis, EPCs, ECFCs, and calcium related signalings, which are not related to the theme of this review. I recommend to simplify this part.
We thank the Referee for this comment. Actually, this review was aimed at introducing with the concept of therapeutic angiogenesis those who were experts in calcium signalling, but not in cardiovascular problems, and, viceversa, at introducing with the basic concept of calcium signalling those who were expert in angiogenesis (e.g. cardiologists and vascular clinicians). This was the rationale for describing with some detail both the mechanisms of vascular growth and the role of Ca2+ signalling in angiogenesis. We, of course, understand your point. On the other hand, Referee #2 congratulated us for the completeness of the information we provided: “This is a well-written and comprehensive work and could represent an important source of information for researchers that work in the field of angiogenesis”. And indeed, suggested us to expand more on the description of VEGF-induced signalling: “In figure 2 the authors represented the Src pathway, but they did not describe it in the text. It would be more useful to provide also the pathway explanation in the text”. We hope that the Referee will understand that we are in between two radically divergent opinions. So, to follow your criticisms, we shortened the length of both the Introduction and Chapter 2 (Vasculogenesis and angiogenesis). But we could not cut more information not to disappoint Referee #1.
Did the authors draw these figures? If not, the authors need to clarify where are these figures adapted from.
Yes, of course, we drew all the figures.
Further perspective should be added in this review.
We thank the Referee for this comment. Future perspective has been described in the Conclusion paragraph.
What is the correlation between TRPV1 and calcium? How TRPV1 stimulate calcium signal in ECs should be summarized.
We thank the referee for this comment. Being mainly located on the plasma membrane, TRPV1 activation results in extracellular Ca2+ entry down the electrochemical gradient at physiological membrane potentials. We have clarified this issue on Lines 299-301 and on Lines 332-336.
The role of TRPV1 in ischemic disease, such as ischemic brain/heart should be illustrated.
We thank the Referee for this comment. To the best of our knowledge, there is no information regarding the role of endothelial TRPV1 in the ischemic brain/heart. However, we addressed this issue on Page 11 by summarizing the cardio- and neuroprotective roles of TRPV1 against the ischemic/reperfusion injury. As mentioned above, we introduced the possibility to investigate whether endothelial TRPV1 activation stimulates revascularization in the ischemic brain/heart as future perspective.
Francesco Moccia, PhD
Laboratory of General Physiology,
Department of Biology and Biotechnology “L. Spallanzani”
University of Pavia,
Via Forlanini 6, 27100, Pavia, Italy.
Tel: 0039 0382 987614.
Fax: 0039 0382 987527.
E-mail: francesco.moccia@unipv.it
Reviewer 2 Report
In this review, Negri and collaborators provided a prospect of Transient Receptor Potential Vanilloid 1 (TRPV1) channel structure, biophysical properties and gating mechanisms, focusing on TRPV1 role in endothelium and, in particular, in the angiogenic process. Moreover, they proposed TRPV1 as a possible target for therapeutic angiogenesis after an ischemic insult.
Generally, this is a well-written and comprehensive work and could represent an important source of information for researchers that work in the field of angiogenesis. However, I have some major comments:
- In figure 2 the authors represented the Src pathway, but they did not describe it in the text. It would be more useful to provide also the pathway explanation in the text.
- There are some evidences in literature that TRPV1 has also an intracellular distribution (https://www.sciencedirect.com/science/article/pii/S0143416017301720; https://onlinelibrary.wiley.com/doi/full/10.1002/jcp.25704). Do you know whether endothelial intracellular TRPV1 could be involved in the angiogenic process?
- As long as we know, temperature has effect on TRPV1 and TRPV4 activation. Is there any evidence that an increase in temperature leads to angiogenesis? And, if so, could this be related to TRPV1?
- In retinal microvascular endothelial cells, TRPV1 assemble with TRPV4. Is there any evidence that this also occurs in ECFCs?
- The authors suggest that light-induced TRPV1 activation in ECFCs is likely related to a photocatalytic reaction. I guess they refer to ROS signalling. Could they expand more on this issue?
I have also some minor comments:
Line 161. Consists IN it’s better consists OF.
Line 171. Also here; consists OF it’s better.
Line 272. PKC is activated by DAG, not PKG.
Line 408. TRPV1 is largely expressed in IN the intimal…. There is 1 IN extra.
Line 431. “it is was”
Line 496. TRPV1 is largely EXPRESSED, not expression.
Line 513. Remove “angiogenesis” from the end of the line.
Line 552. Apart FROM, it’s missed.
Line 776. Do not ALLOW, not allowing.
Author Response
Dear Referee #2,
we thank you for the nice comment on our manuscript entitled: Endothelial TRPV1 as an emerging molecular target to promote therapeutic angiogenesis recently submitted for publication as a Review article in Cells.
We believe that your comments did improve the quality of the manuscript and we are pleased for the attention you paid to our work. We amended the manuscript according to your indications and addressed all the criticism you raised. More specifically:
- In figure 2 the authors represented the Src pathway, but they did not describe it in the text. It would be more useful to provide also the pathway explanation in the text.
We thank the referee for this comment. We implemented the text with the Src pathway description on Lines 199-203.
- There are some evidences in literature that TRPV1 has also an intracellular distribution (https://www.sciencedirect.com/science/article/pii/S0143416017301720; https://onlinelibrary.wiley.com/doi/full/10.1002/jcp.25704). Do you know whether endothelial intracellular TRPV1 could be involved in the angiogenic process?
Thank you for the interesting comment. We know that TRPV1 is intracellularly expressed in several cell types, including rat nociceptive neurons (10.1074/jbc.M310891200) and some prostate (https://doi.org/10.1016/j.bbamcr.2016.09.013) and breast (10.2147/BCTT.S170208) cancer cell lines. One recent investigation by O’Leary and coworkers disclosed the cytosolic distribution of TRPV1 also in bovine retinal microvascular endothelial cells, but it did not assess whether intracellular TRPV1 was involved in angiogenesis. This issued has now been addressed on Lines 336-340 and the references kindly suggested by the Referee were added to the text.
- As long as we know, temperature has effect on TRPV1 and TRPV4 activation. Is there any evidence that an increase in temperature leads to angiogenesis? And, if so, could this be related to TRPV1?
We do thank the referee for this comment. As a pioneering paper by Nilius group suggested that TRPV4 could enable mouse aortic endothelium to detect moderate increases in temperature, thereby controlling vascular tone and permeability. Moreover, a more recent paper suggested that also TRPV1 could sense heat increases to over 40°C in human corneal endothelial cells (Mergler et al., Exp Eye Res. 2011 Nov;93(5):710-9). To address your criticism, we implemented the manuscript by adding paragraph 5.4: Is there a role for TRPV1 in heat-induced angiogenesis?
- In retinal microvascular endothelial cells, TRPV1 assemble with TRPV4. Is there any evidence that this also occurs in ECFCs?
We thank the referee for the question. It is known from a long time that also TRPV4 is expressed in ECFCs. According to Lodola and coworkers, it is unlikely that TRPV4 assembles with TRPV1 in these cells (https://doi.org/10.1126/sciadv.aav4620). For instance, photostimulation-induced membrane depolarization in ECFCs plated on P3HT was prevented by the TRPV1-specific inhibitor capsazepine, but not by the TRPV4-specific inhibitor RN-1734 (https://doi.org/10.1126/sciadv.aav4620). In addition, the photostimulation mechanism based on photoelectrochemical reactions also contribute to explain why TRPV4 is not sensitive to optical stimulation and, therefore, does not assemble with TRPV1. Indeed, TRPV4 is inhibited rather than activated by pH variation induced by photo-generated electrons (https://doi.org/10.1126/sciadv.aav4620). Furthermore, a previous investigation of our group revealed that the Ca2+ signal mediated by direct TRPV4 stimulation with GSK1016790A (GSK) was not sensitive to TRPV1 inhibition with capsazepine (10.1002/jcp.24686). These issues were not addressed on Lines 742-752 and on Lines 668-672.
- The authors suggest that light-induced TRPV1 activation in ECFCs is likely to be due to a photocatalytic reaction. I guess they refer to ROS signalling. Could they expand more on this issue?
Thank the reviewer for the comment. Your hypothesis is right, and we added a more detailed description in the text (Lines 742-752).
We therefore hope you will now regard the manuscript worth of being published on Cells.
Sincerely,
Francesco Moccia
Francesco Moccia, PhD
Laboratory of General Physiology,
Department of Biology and Biotechnology “L. Spallanzani”
University of Pavia,
Via Forlanini 6, 27100, Pavia, Italy.
Tel: 0039 0382 987614.
Fax: 0039 0382 987527.
E-mail: francesco.moccia@unipv.it
Round 2
Reviewer 1 Report
The authors have addressed all my concerns
Author Response
We thank the reviewer for his/her kind comments!